# Unifying Node Labels, Features, and Distances for Deep Network Completion

**DOI:** 10.3390/e23060771

**Published:** 2021-06-18

**Authors:** Qiang Wei, Guangmin Hu

**Affiliations:** 1School of Information and Communication Engineering, University of Electronic Science and Technology of China, Chengdu 611731, China; hgm@uestc.edu.cn; 2National Key Laboratory of Science and Technology on Blind Signal Processing, Chengdu 610041, China

**Keywords:** network completion, graph convolutional network, node label, node feature, node distance

## Abstract

Collected network data are often incomplete, with both missing nodes and missing edges. Thus, network completion that infers the unobserved part of the network is essential for downstream tasks. Despite the emerging literature related to network recovery, the potential information has not been effectively exploited. In this paper, we propose a novel unified deep graph convolutional network that infers missing edges by leveraging node labels, features, and distances. Specifically, we first construct an estimated network topology for the unobserved part using node labels, then jointly refine the network topology and learn the edge likelihood with node labels, node features and distances. Extensive experiments using several real-world datasets show the superiority of our method compared with the state-of-the-art approaches.

## 1. Introduction

### 1.1. Background

Network structures, such as social networks, web graphs, and communication networks, are important to the functioning of complex systems [1,2]. Usually, a complete network structure is a crucial prerequisite for downstream tasks, including node classification and link prediction [3,4,5]. However, real-world networks tend to be partially observed, with nodes and edges missing due to insufficient resources and privacy protection [3,6,7]. Social networks, such as Twitter and Facebook, have restrictions for crawlers, which makes it impossible for third-party aggregators to collect complete network data. Similarly, when an internet scientist probes the route topology using traceroute, they cannot obtain the structure behind the non-cooperative routes [8]. Thus, the collected network structure is often incomplete, which creates difficulties for downstream analysis.

Our work is motivated by learning the structure of communication networks from passive measurements. In a military context, we may wish to analyze a foreign network inconspicuously [9]. One feasible way is to monitor the packet traffic between the target network and our controlled networks. In an internet reconstruction context, we may wish to obtain a map of networks that have connections with our hosted one for routing strategy optimization [10,11]. In both scenarios, we place passive traffic collectors, and we can collect the user profile (e.g., IP address) [12], hop distance (via TTL) [9,13], and label (via a community) [2] through continuous passive monitoring of the communication, starting from the target network [14]. Passive monitoring provides rich information but there are two limitations: (1) it is often impractical to collect edges or relationships between nodes within the target networks, as their traffic does not pass through our collectors; and (2) there is little control over which targets are measured and, therefore, some data are invariably missing [14].

The difficulty of collecting edges in target networks leads us to the network completion (NC) problem [7]: given a partially observed network structure H from a underlying network G, the goal is to infer the missing unobserved part Z. Network completion can significantly alter our estimates of network-level statistics and node-level structure. In [7], both nodes and edges were missing in Z; however, the nodes are known and only the edges are missing in our scenarios. We consider this setting as a specific network completion problem [15]. Solving the problem is particularly challenging, as all edges in Z are missing. While intuitively similar, NC is fundamentally different from the related well-studied link prediction problem [16]. Edges are missing at random in the link prediction; by contrast, the nodes and edges are missing as a whole in NC, which makes traditional link prediction algorithms unsuitable for NC.

As aforementioned, we focused on solving the specific network completion problem, where node information is known and only edges are missing in Z. The information of nodes, such as labels, features, and distances, is available in our settings [15,17]. It is proven that the side information of nodes is strongly correlated with the underlying network structure [18,19,20,21,22]. Therefore, it is beneficial to integrate side information for NC.

### 1.2. Related Methods

**Model-Based NC:** Kim and Leskovec developed KronEM [7], an expectation maximization approach combined with the Kronecker graphs model. They designed a scalable, metropolized Gibbs sampling approach for the estimation of model parameters, as well as inference of the missing part Z. KronEM suffers from three problems [16]: (1) only network topology is considered and side information of nodes is ignored; (2) not all real-world networks follow the Kronecker model; and (3) its speed and performance are not yet satisfactory.

**Node-Similarity-Based NC:** Other than the missing edges, the node identification and features in *G* may be available in real settings. Node-similarity-based network completion methods leverage the similarities between node features to infer Z. Matrix completion with decoupled transduction (MC-DT) [15] decouples the completion from transduction to effectively exploit the similarity information. Furthermore, joint node clustering and similarity learning (JCSL) [23] handles the situation, where the node features may be partially missing by computing the node similarities at the cluster level, then jointly co-factorizing the observed adjacency matrix with the cluster-based similarities. However, MC-DT and JCSL have two shortcomings: (1) choosing an appropriate similarity metric is a prerequisite but challenging in practice [24]; and (2) the similarity matrix is exploited linearly, which cannot reflect the nonlinearity between node features and network structures.

**Network Structure Learning:** Instead of using a similarity graph based on the initial features, another approach in this category is to learn a network structure. Graph neural networks (GNNs) have become the standard toolkit for learning from networks [25,26]. However, most GNNs are designed for a relatively complete network structure, which makes them unsuitable for the network completion problem. Franceschi et al. sampled graph structures from a learnable fully connected structure and employed a bi-level optimization setup for simultaneously learning the GNN parameters and the structure [27]. Chen et al. proposed an iterative method to search for a hidden graph structure that augments the initial graph structure toward an optimal graph for (semi-)supervised prediction tasks [28]. Yu et al. introduced a GCN-based graph revision module for predicting missing edges and revising edge weights via joint optimization [29]. Hao et al. embedded the graph nodes into latent space, and then computed an embedding vector for the unobserved node, with attributes compared to another node’s embedding for link prediction [30]. Shin et al. presented Edgeless-GNN for attributed network embedding with edgeless nodes by utilizing a k-nearest neighbor graph based on the similarity of node attributes. However, the existing network structure learning works ignore node labels and distances.

**Deep Generative Graph Models:** Recent advances in deep generative graph models have furthered progressed in network completion. Inspired by GraphRNN [31], DeepNC infers the missing parts of a network via deep learning for solving NE-NC [16]. DeepNC first learns a likelihood over the edges via an RNN-based generative graph model by using structurally similar graphs as training data, then inferring the missing parts by applying an imputation strategy for the missing data. Similar to KronEM, DeepNC does not take advantage of the side information of nodes. In addition, DeepNC requires structurally similar graphs for training, which are often difficult to collect in reality; for example, route-level networks are hard to obtain as privacy protection within autonomous systems [32]. Graphite [33] parameterizes variational autoencoders with GNN and uses an iterative graph-refinement strategy inspired by low-rank approximations for decoding. G-GCN [34] produces a unified generative graph convolutional network that learns node embeddings for all nodes by sampling graph generation sequences constructed from H. Graphite and G-GCN are designed for edgeless NC, but neither of them considers node label or distance information, which leaves room for further improvement.

### 1.3. Present Work

In this work, we address the challenge of network completion using side information. We propose a node-label-, feature-, and distance-based network completion method (LFD-NC), a novel unified deep graph convolutional network that infers the missing edges by leveraging the node label, feature, and distance information. Specifically, we first construct an initial network topology for Z by node labels using a stochastic block model, and then we adopt GNN to obtain a refined estimation of Z using node features; after that, distance constraints are used for pruning the refined network. Lastly, we learn a joint distribution over Z by iteratively performing refinement and pruning.

The contributions of this paper are threefold:We formalize NC with side information as a graph refinement problem;We propose LFD-NC, a deep graph convolutional-network-based completion method by unifying the observed structure with node label, feature, and distance information;We validate LFD-NC through extensive experiments on several real-world networks.

## 2. Methods

We begin by introducing the network completion problem with side information; then, we propose LFD-NC, a deep graph convolutional-network-based algorithm, to solve the problem.

### 2.1. Problem Formulation

We assume that there is a true undirected and unweighted network GV, E, X, Y, where V=v1,v2,…vN denotes the node set with V =N, E⊂V×V denoting the edge set, X∈ℝN×F is the feature matrix for each node in V, and Y∈1,2,…,CN represents the node labels of C classes or communities.

We consider the NC task with node side information over G, where only a part of the topology of G is observed. As illustrated in Figure 1, we have complete knowledge of X, Y, as well as an observed induced subgraph $O\left(V_{O},\;E_{O}\right)$ of G. We denote $A_{O}\;=\left[w_{O}\left(u,v\right)\right]\;\in {\left\{0,1\right\}}^{N\times N}$ as the observed adjacency matrix, where $w_{O}\left(u,v\right)=1$ if $\left(u,v\right)\;\in E_{O}$; otherwise, $w_{O}\left(u,v\right)=0$. We also know the shortest-path distance set $D=\{\left(u,v,d_{uv}\right)|u\in V_{OD},v\in V_{ZD}\}$ for some node pairs between $V_{OD}\subseteq V_{O}$ and $V_{ZD}\subseteq V\backslash V_{O}$, and we denote $D_{u}=\{v|\left(u,v,d_{uv}\right)\in D\}$ as the observed destination nodes set from source node u. Let $V_{Z}=V\backslash V_{O}$ be the node set of the unobserved topology, and let $M_{Z}=\left\{V_{O}\times V_{Z}\right\}\;\cup \;\left\{V_{Z}\times V_{Z}\right\}$ represent all possible edges with at least one endpoint in $V_{Z}$. The task of NC is to infer the missing edges and non-edges in the unobserved part $Z\left(V_{Z},\;E_{Z}\right)$ where $E_{Z}\subset M_{Z}$.

It is worth mentioning that the observed distance set D considered here is retrieved from passive measurements; therefore, we cannot control the observed shortest-path destination nodes $D_{u}$ for a given source node $u\in V_{OD}$. In this paper, we further assume that for any source node $u\in V_{OD}$, the destination node set $D_{u}$ satisfies $D_{u}\subset V_{Z}$; that is, we cannot detect all the shortest-path distances from u. Although D indicates deterministic edges and non-edges in Z (see Section 2.5), we treat them as unobserved for the simplicity of the NC problem definition.

Let us model $E_{Z}$ through the following probability distribution:(1)$$E_{Z}∼P_{G}(E_{Z}|O,X,Y,D,\Theta ),$$
which is parameterized by Θ. Then, the objective of E-NC is to find the most likely configuration of $E_{Z}$.

### 2.2. Overview of LFD-NC

LFD-NC is motivated by the fact that network topology, node labels, and node features are correlated. Thus, we aimed to find the optimized network topology $E_{Z}$ that is most consistent with the observed O, X, Y, and  D. As there is no closed form for the posterior distribution $P_{G}(E_{Z}|O,X,Y,D,\Theta )$ in Equation (1), LFD-NC models it by iteratively refining the network topology. It requires four phases in our computing framework: label-based topology initialization, edge probability learning, distance pruning, and topology refinement. The LFD-NC architecture is shown in Figure 2.

.

In the label-based topology initialization phase, LFD-NC computes an estimated topology $G_{L}$ for Z, using only node label information Y. Then, in the edge probability learning phase, we take $G_{L}$, node feature X and node label Y as the inputs to learn a new edge probability $P_{X}\left(u,v\right)$ for $\left(u,v\right)\in M_{Z}$, which leads to a better topology $G_{X}.\;$ Notably, learning $P_{X}$ is a typical link-prediction task, where standard methods such as GNN [35,36] or improved methods with label propagation [37,38,39] can be directly applied as aforementioned.

We then prune edges that should not exist on the basis of node distance constraints D. Lastly, we refine the estimated topology $G_{X}$ by iteratively performing edge probability learning and distance pruning.

From a node-embedding perspective, we first map each node in G to a low-dimensional vector in LFD-NC’s first two phases, then we learn a pairwise decoder to predict the edges and non-edges in Z in the edge probability learning phase. Lastly, we refine the network topology in the distance pruning phase, and recalculate the node embeddings correspondingly using topology refinement.

Next, we elaborate the details of each step.

### 2.3. Label-Based Topology Initialization

NC suffers from the cold start problem [15,34], i.e., there are no prior connections for the nodes in Z. Therefore, standard GNN methods, such as GCN [35] and GAT [40], cannot be directly applied for NC since the missing edges block message passing and aggregation between O and Z.

We present node-label-based topology initialization to overcome the cold start problem in NC, which is motivated by two insights. Firstly, the community structure is positively correlated with the underlying network topology [21]. Most complex networks show a community structure, i.e., blocks of nodes that have a high density of edges within them, and a lower density of edges between them. Community structure is often detected on the basis of the known underlying network topology. Here, we take the opposite direction and treat the community structure as a good initial estimation of the unobserved network topology. Secondly, the label information can be integrated to improve the performance of node embedding. It is proven that unifying label propagation and GNN overcomes the over- or under-smoothing issue of GNN [37,38,41]. In this paper, we treat the known labels Y as the optimized result of the label propagation procedure, and then find the corresponding topology.

We initialize edges in MZ using the stochastic block model (SBM) [3,42,43], which is widely used to model communities in complex networks by modulating the intra- and extra-block connections. Specifically, the edge probability in MZ is determined by the following:(2)PLu,v =pYu,Yv,
where pYu,Yv∈0,1C×C is the probability of an edge between communities. The estimated edge weight in MZ is calculated as follows:(3)wLu,v =αLPLu,v,
where αL is a parameter that controls the strength of PL in the estimated graph GL. If we set wLu,v =0 for u,v∈VO, and denote WL= wLu,v ∈ℝN×N as the SBM-estimated matrix, then the adjacency matrix of GL can be represented by the following:(4)AGL=AO+WL.

We show that $A_{G_{L}}$ can be the underlying topology for label propagation. The label propagation procedure in iteration k can be formulated as follows [37]:(5)$$Y^{\left(k+1\right)}={\tilde{D}}^{-1}A_{G}Y^{\left(k\right)},$$
where $A_{G}=\;\left[a_{ij}\right]\;\in {\left[0,1\right]}^{N\times N}$ is the partial unknown adjacency matrix of G, and $\tilde{D}$ is a diagonal matrix with ${\tilde{D}}_{ii}=\sum_{j}{\tilde{A}}_{ij}$. We have $Y^{\left(k+1\right)}=Y^{\left(k\right)}=Y$ in our settings; hence Equation (5) holds when $A_{G}=A_{G_{L}}$, which indicates that $A_{G_{L}}$ is a valid solution of Equation (5). Therefore, $A_{G_{L}}$ is consistent with the label propagation.

### 2.4. Edge Probability Learning

Edge probability learning aims to obtain a better network topology from $G_{L}$, X and Y, and we treat it as a link prediction task. After the label-based topology initialization phase, we directly exploit network embedding techniques to find the proper function f as follows:(6)$$H=f\left(A_{G_{L}},X,\;Y\right),$$
where H is the node embedding matrix.

Many existing methods can be used to obtain H [35,36,37,38,39]. In this paper, we adopt GCN [35] to model f. In GCN, the hidden representations for each layer can be obtained by the following: (7)H(l+1)=σD˜−12A˜ D˜−12HlWl,
where A˜=AGL+IN is the adjacency matrix of GL with added self-loops, IN is the identity matrix, D˜ is a diagonal matrix with D˜ii=∑jA˜ij, Wl∈ℝFl×Fl+1 is a layer-wise learnable weight matrix, σ· denotes an activation function such as ReLU [44], and Hl is the node embedding of layer l with H0=[X|Y], where [X|Y] is a concatenation of X and one-hot label indicators.

We consider a two-layer GCN as our forward model, and the final embedding is the following:(8)$$H=\hat{A}\;\mathrm{ReLU}(\hat{A}[X|Y]W^{\left(0\right)})W^{\left(1\right)},$$
where $\hat{A}={\tilde{D}}^{-\frac{1}{2}}\tilde{A}\;{\tilde{D}}^{-\frac{1}{2}}$. The weight matrices $W^{\left(0\right)}$ and $W^{\left(1\right)}$ are calculated by minimizing the cross-entropy errors of labeled edges in O:(9)$$L_{\mathrm{cross}-\mathrm{entropy}}=-\sum_{\left(u,v\right)\in E_{O}}\log\left(P_{X}\left(u,v\right)\right)-\sum_{\left(u,v\right)\notin E_{O}}\log\left(1-P_{X}\left(u,v\right)\right).$$

Due to the sparse nature of real-world networks, there are only a small number of edges in all node pairs; thus, we generate $\left|E_{O}\right|$ non-edges via random sampling.

The edge probability in $M_{Z}$ then takes the following simple form:(10)$$P_{X}\left(u,v\right)=\mathrm{sigmoid}\left(H_{u}H_{v}^{T}\right).$$

As a function Equation (10), we provide the realization of Equation (1) as follows:(11)$$P_{G}(E_{Z}|O,X,Y,D,\Theta )=\prod_{\left(u,v\right)\in E_{Z}}P_{X}\left(u,v\right)\prod_{\left(u,v\right)\notin E_{Z}}(1-P_{X}\left(u,v\right)).$$

We denote PX=pXu,v ∈ℝN×N as the link likelihood matrix, and set pXu,v=0 for u,v∈VO; then, the adjacency matrix of GX can be represented by the following:(12)AGX=AO+PX.

### 2.5. Distance Pruning and Topology Refinement

Distance pruning and topology refinement aim to further improve the performance of node embedding. The distance constraint D indicates the existence of edges and non-edges between some node pairs, whereby clamping the edge probability of these node pairs leads to a clearer network topology $G_{D}$. Then, we take $G_{D}$ instead of $G_{L}$, and repeat the edge probability learning process to gradually refine the node embedding matrix H.

Given the distance constraint D, we may calculate two deterministic sets: an edge set $E_{D}\subset M_{Z}$ and a non-edge set $E_{D}^{\emptyset }\subset M_{Z}$. The calculation is based on Observation 1 and Observation 2 [11].

***Observation*** ***1.***
*For any given nodes u, v and w in an undirected and unweighted graph $G\left(V,E\right)$, if $|d_{uv}-d_{uw}|\geq 2$, then $\left(w,v\right)\notin E$ holds.*


***Observation*** ***2.***
*Let $L_{u}^{i}=\{v|d_{uv}=i,v\in N\}$ be the sets of nodes with the same distance i from u. For two given nodes $v\in L_{u}^{i}$ and $w\in L_{u}^{i+1}$, if for any node $x\in L_{u}^{i}\backslash \left\{v\right\}$, $\left(x,w\right)\notin E$, then $\left(v,w\right)\in E$ holds.*


Note that Observation 2 needs u observed distances to all the other nodes in G, which cannot be met under our assumption. Therefore, $E_{D}$ only contains the observed direct neighbors of the distance monitor nodes $V_{OD}$.

After the calculation of $E_{D}$ and $E_{D}^{\emptyset }$, we clamp the probability of edges in $E_{D}$ to 1, and the probability of non-edges in $E_{D}^{\emptyset }$ to 0. Let $M_{non-edge}=\left[m\left(w,v\right)\right]\in {\left\{0,1\right\}}^{N\times N}$ denote the non-edge mask matrix where $m\left(w,v\right)=1$ if $\left(w,v\right)\in E_{D}^{\emptyset }$. Then, the distance pruning process can be represented as follows:(13)$$A_{G_{X}}\left[M_{non-edge}\right]=0.$$

Then, we assign the adjacency matrix $A_{G_{D}}$ of $G_{D}$ as the masked $A_{G_{X}}$. We ignore $E_{D}$ in LFD-NC as $\left|E_{D}\left|\ll \right|E_{D}^{\emptyset }\right|$.

We summarize our algorithm in Algorithm 1.

The time complexity of LFD-NC is the same as that of GCN. The complexity of line 1 in Algorithm 1 is O(|MZ|). In GCN, it is usually satisfied that F>F1≥F2; thus, the complexity of line 3 is ON2F+NF2. The complexity of lines 4 and 5 is O(|MZ|). Since |MZ|<N2, the total complexity of LFD-NC is dominated by GCN.
**Algorithm 1.** LFD-NCInput: node features X, non-edge mask Mnon−edge, observed graph matrix AO, SBM estimated matrix WL, and topology refinement round R.Output: estimated graph AGD.1.AGL=AO+WL//label-based topology initialization by Equation (4)2.forrin 1,2,…,R//topology refinement3.H=fAGL,X//node embedding by Equation (6)4.AGX=AO+PX//link prediction by Equation (12)5.AGXMnon−edge =0//distance pruning by Equation (13)6.AGL=AGD=AGX//update AGL7.*end for*
8.outputAGD


## 3. Experiments

We conducted a series of experiments. First, we evaluated the performance of LFD-NC and compared it with the state-of-the-art network completion methods. Then, we analyzed the impacts of the four phases.

### 3.1. Experimental Settings

#### 3.1.1. Datasets

We evaluated the performance of LFD-NC on eight real-world network datasets. The details of the eight datasets are presented below.

Actor is a film actor network [45]. Each node corresponds to an actor, and the edge between two nodes denotes co-occurrence on the same Wikipedia page. Node features correspond to some keywords in the Wikipedia pages. Nodes are classified into five categories in terms of words in the actor’s Wikipedia entry.

Cornell, Texas, and Wisconsin are three small local web networks from WebKB [45], which is a webpage dataset collected by Carnegie Mellon University from computer science departments at various universities. Nodes and edges represent web pages and hyperlinks, respectively. The feature of each node is the bag-of-words representation of the corresponding page, and the label of each node is manually classified into five categories: student, project, course, staff, and faculty.

Cora, Citeseer, and PubMed are three classic citation networks [46]. Nodes represent scientific papers, and edges represent citation relationships. Node features correspond to the bag-of-words representation of the paper and the label of each node represents one of the academic topics.

WikiCS is a Wikipedia web network [47]. It consists of nodes corresponding to computer science articles, with edges based on hyperlinks. Node features are derived from the text of the corresponding articles. Nodes are labeled into 10 classes representing different branches of the field.

The dataset statistics are shown in Table 1. We only focused on the largest connected component for each network in this paper.

#### 3.1.2. Baselines and Evaluation Metrics

The baselines were chosen from five different types of network completion algorithm for comparison: SBM [42] only uses node labels, whereby the symmetric C×C matrix of edge probability $\left[p_{Y\left(u\right),Y\left(v\right)}\right]$ is estimated from the observed O;KronEM [7] only uses the network graph structure and ignores node features, node labels, and node distances;MC-DT [15] employs both the pairwise similarity of node features and the network graph structure, as well as ignores node labels and node distances. The similarity information is utilized by matrix factorization in a linear way;MLP-NC [48] considers node features and the network graph structure, as well as ignores node labels and node distances. Unlike MC-DT, MLP-NC directly learns a non-linear similarity metric;G-GCN [34] also considers node features and the network graph structure, as well as ignores node labels and node distances. Unlike MLP-NC, G-GCN adopts a generative graph convolution model.

We treated the prediction of missing edges in Z as a binary classification, and we evaluated the performance of LFD-NC on the basis of two metrics: the area under the ROC curve (AUC) and average precision (AP). We randomly sampled equal numbers of negative and positive edges when evaluating AUC and AP.

#### 3.1.3. Implementation Details

We generated $O\left(V_{O},\;E_{O}\right)$ by breadth-first search (BFS) traversal from a randomly selected node and split $E\backslash E_{O}$ equally for validation and testing. Before training, node features X were normalized, and then extended by concatenating the one-hot encoding of Y for a fair comparison with MC-DT, MLP-NC and G-GCN.

We adopted the realization of KronEM in SNAP [49] and kept all parameters as the default, except for the initial Kronecker matrix, which we set to $\left[0.9,\;0.7;\;0.7,\;0.2\right]$ for symmetry. We implemented MC-DT, using the SciPy Python library, while the feature similarity matrix was computed by the cosine metric, and we set the number of eigenvectors s to 20. We executed G-GCN from the officially released code, and kept all parameters as the default, except for $F^{\left(1\right)}$ and $F^{\left(2\right)}$. We implemented MLP-NC and LFD-NC in PyTorch and open-sourced them at https://github.com/weiqianglg/LFD-NC.

For LFD-NC, we constructed D by randomly selecting a fixed number of source nodes for $V_{OD}$; then, we randomly split $V_{Z}$ into $\left|V_{OD}\right|$ disjoint subsets, where each subset corresponded to one source node. The topology refinement round was set to R=4. We adopted the planted partition model in SBM initialization, and we fixed $p_{Y\left(u\right),Y\left(v\right)}=1$ if nodes u, v had the same label; otherwise, $p_{Y\left(u\right),Y\left(v\right)}=0$ in Equation (2). For G-GCN, MLP-NC, and LFD-NC methods, we set the hidden dimension as $F^{\left(1\right)}=64$ and the final dimension as $F^{\left(2\right)}=32$, and we used the Adam optimizer with a learning rate of 0.01 to train the three deep-learning models for all the datasets. We also applied an early stop strategy over AUC for the validation set, with the patience set to 20 epochs.

### 3.2. Completion Performance

#### 3.2.1. Comparison with State-of-the-Art Methods

Table 2 and Table 3 summarize the comparison results for the eight real-world network datasets. We kept $|V_{OD}|=16$ and $\left|E_{O}\right|/\left|E\right|=0.8$, and we set $\alpha_{L}=10^{-4}$ for Cornell, Cora, Pubmed, and WikiCS, $\alpha_{L}=6\times 10^{-2}$ for Actor and Texas, $\alpha_{L}=8\times 10^{-2}$ for Wisconsin, and $\alpha_{L}=1\times 10^{-1}$ for Citeseer. The mean and the confidence interval of AUC and AP were measured by 10 random BFS samplings. LFD-NC outperformed all other comparative models in the eight datasets. These results demonstrate that incorporating node label and distance constraints into GNN models significantly improves the solution of an NC problem.

KronEM uses only the observed network topology for completion, and SBM uses only node labels, which led to them performing the worst. MC-DT, MLP-NC, and G-GCN performed similarly to each other in general, but they consider neither node labels nor distance constraints; therefore, LFD-NC outperformed almost all of them. For example, LFD-NC achieved about 21.9–23.2% AUC and 16.1–22.5% AP absolute improvements, compared with the second-best methods in Cornell and Texas. These improvements were contributed by both node label information and distance constraints; in Cora and Citeseer, LFD-NC achieved 5.5–7.3% AUC and 5.2–8.4% AP absolute improvements, compared with the second-best methods. These improvements were mainly contributed by distance constraints. We systematically studied the impact of node labels and distance constraints, as presented below.

#### 3.2.2. Impact Analysis of Node Labels

Node labels are integrated into the initial topology by the SBM-estimated matrix, which is controlled by the parameter $\alpha_{L}$, as shown in Equations (3) and (4). The SBM estimation enhances the connections for nodes in the same class, but it creates noisy edges that do not actually exist. Therefore, we need to properly set the parameter $\alpha_{L}$.

Figure 3 shows the impact of parameter $\alpha_{L}$ on AUC and AP in Actor and Cora. The importance of $\alpha_{L}$ clearly varied with the dataset. A properly chosen $\alpha_{L}$ produced 5% absolute AUC and AP improvements, compared with $\alpha_{L}=0$ in Actor, whereas there was no improvement in Cora.

The effectiveness of the label-based topology initialization was affected by the correlation between the node features and edges. When $\alpha_{L}$ was close to zero, LFD-NC degenerated into a MLP model, which did not use label information; when $\alpha_{L}$ became large, more noisy edges were added into the initial topology $A_{L}$, which eventually degraded the performance. Therefore, a small $\alpha_{L}$ produced good results in Cora, where the edge likelihood was mainly determined by the node features; however, in Actor, neither small nor large $\alpha_{L}$ produced the best results.

#### 3.2.3. Impact Analysis of Distance Constraints

Figure 4 presents the impacts of the distance constraints on AUC and AP. A higher deterministic edge rate $\left|E_{D}^{\emptyset }\left|/\right|M_{Z}\right|$ achieved about 5.3–14.4% AUC and 5.5–12.6% AP absolute improvements, compared with zero monitors ($|E_{D}^{\emptyset }\left|/\right|M_{Z}|=0$). Distance pruning restricted the probability of edges in Z and reduced the uncertainty of the estimated topology; therefore, the AUC and AP of LFD-NC increased with $\left|E_{D}^{\emptyset }\left|/\right|M_{Z}\right|$.

#### 3.2.4. Ablation Study

LFD-NC solves the NC problem in four sequential phases; here, we performed an ablation study in LFD-NC on Actor and Texas datasets, as shown in Table 4. Compared with the standard LFD-NC, removing the label-based topology initialization phase resulted in the highest decrease in AUC and AP (6%); removing the topology refinement phase resulted in the highest decrease in AUC and AP (11%); and removing the distance pruning phase resulted in the highest decrease in AUC and AP (>26%). The experiments show that the four phases are all necessary for LFD-NC, and the distance pruning phase is particularly important.

## 4. Conclusions and Future Work

We presented LFD-NC, a unified deep graph convolutional network for network completion. LFD-NC integrates node label, feature, and distance information through a graph refinement framework. Experiments on eight datasets demonstrated that our model outperforms state-of-the-art baseline methods.

Our work can be easily extended to directed graphs and multigraphs. To treat directed graphs, the only necessary change is to perform directed node embedding in the edge probability learning phase. To treat multigraphs, we take each type of relationship individually, and then combine all the inferred results to achieve a final completion.

We note three possible directions for future work. Firstly, our proposed model assumes that node labels Y and node features X are completely known. An interesting direction would be to perform completion when parts of X and Y are missing and noisy. Secondly, LFD-NC has a long training time for large-scale networks; thus, reducing the time complexity is also a possible future research direction. Thirdly, we integrated distance constraints from randomly selected monitors; therefore, an effective monitor placement strategy can be designed for further performance improvement.

## Figures and Tables

**Figure 1 entropy-23-00771-f001:**
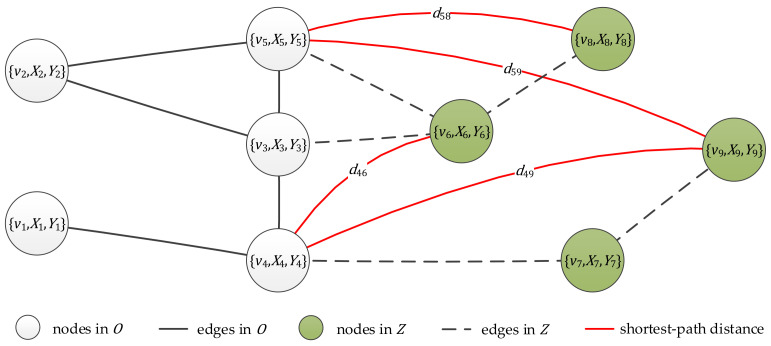
The NC problem with node side information. All nodes $\left\{v_{1},v_{2},\ldots ,v_{9}\right\}$, node feature matrix X, and node label vector Y are known. Edges (black solid lines) in O are observed, but edges (black dashed lines) in the unobserved part Z are missing. Some shortest-path distances (red solid lines) between O and Z are known in addition. To solve NC is to infer the missing edges and non-edges in Z, e.g., edge $\left(v_{3},\;v_{6}\right)$ and non-edge $\left(v_{6},\;v_{9}\right)$.

**Figure 2 entropy-23-00771-f002:**
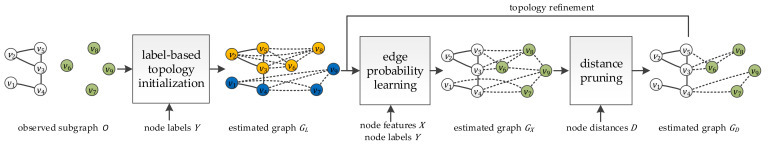
Overview of our LFD-NC. Dashed lines are learned edge probabilities. Nodes with the same color have the same label in $G_{L}$.

**Figure 3 entropy-23-00771-f003:**
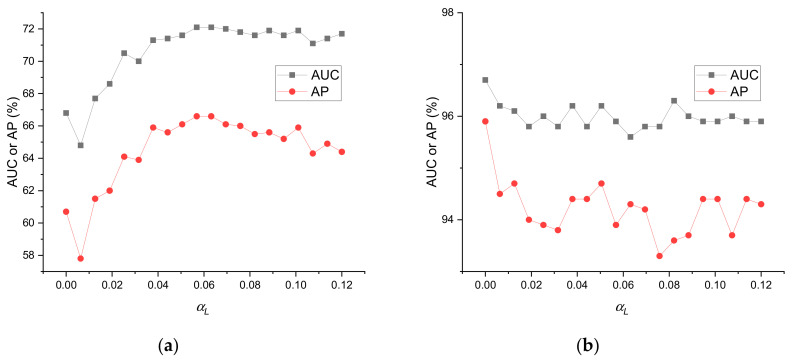
Impacts of the parameter $\alpha_{L}$ on AUC and AP with $\left|E_{O}\right|/\left|E\right|=0.8$ for (**a**) Actor and (**b**) Cora.

**Figure 4 entropy-23-00771-f004:**
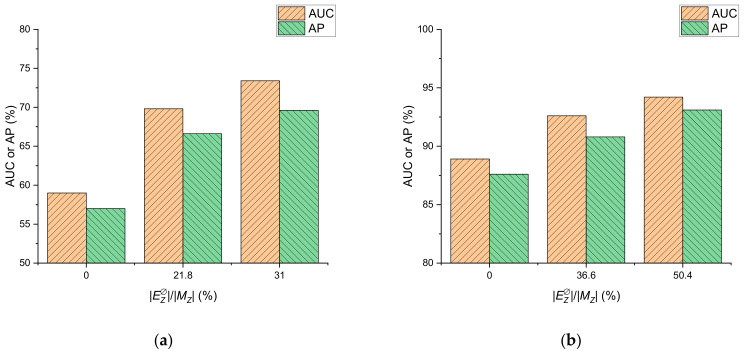
Impacts of the distance monitor number on AUC and AP with $\left|E_{O}\right|/\left|E\right|=0.5$ for (**a**) Actor and (**b**) Cora.

**Table 1 entropy-23-00771-t001:** Dataset statistics.

Dataset	Nodes	Edges	Classes	Features
Actor	7600	33,544	5	931
Cornell	183	295	5	1703
Texas	183	309	5	1703
Wisconsin	251	499	5	1703
Cora	2708	5429	7	1433
Citeseer	3327	4732	6	3703
PubMed	19,717	44,338	3	500
WikiCS	11,701	216,123	10	300

**Table 2 entropy-23-00771-t002:** Comparison of test set AUC score (%) with state-of-the-art methods. The best results are marked in bold.

Method	Actor	Cornell	Texas	Wisconsin	Cora	Citeseer	PubMed	WikiCS
SBM	49.7 ± 0.6	45.1 ± 5.8	65.2 ± 8.0	52.7 ± 7.0	88.9 ± 1.5	80.7 ± 2.0	76.1 ± 1.6	83.1 ± 0.8
KronEM	54.0 ± 1.2	52.0 ± 12.3	59.3 ± 9.7	51.5 ± 9.2	50.9 ± 1.4	49.3 ± 2.1	57.6 ± 1.7	62.2 ± 2.9
MC-DT	50.6 ± 0.7	48.9 ± 9.4	58.8 ± 10.5	72.7 ± 2.9	91.6 ± 1.6	87.7 ± 1.6	89.4 ± 0.8	92.3 ± 0.9
MLP-NC	51.6 ± 1.1	43.1 ± 9.1	41.8 ± 9.4	66.7 ± 8.0	90.1 ± 0.9	85.6 ± 1.6	88.6 ± 1.2	92.3 ± 1.2
G-GCN	50.4 ± 0.7	53.8 ± 4.5	38.9 ± 7.8	60.0 ± 6.0	93.2 ± 0.2	88.7 ± 0.2	88.7 ± 0.2	90.7 ± 0.5
LFD-NC	**72.1** ± 1.0	**85.7** ± 2.3	**88.4** ± 2.8	**89.4** ± 4.2	**97.1** ± 0.6	**96.0** ± 0.7	**93.7** ± 0.9	**92.5** ± 0.9

**Table 3 entropy-23-00771-t003:** Comparison of test set AP score (%) with state-of-the-art methods. The best results are marked in bold.

Method	Actor	Cornell	Texas	Wisconsin	Cora	Citeseer	PubMed	WikiCS
SBM	49.9 ± 0.5	49.4 ± 3.9	68.7 ± 8.3	56.7 ± 4.5	85.8 ± 2.1	79.2 ± 1.8	71.3 ± 1.7	81.6 ± 0.9
KronEM	53.0 ± 1.3	56.0 ± 11.4	59.9 ± 7.6	55.0 ± 7.9	51.5 ± 1.4	50.0 ± 1.6	55.3 ± 1.6	59.9 ± 2.5
MC-DT	51.1 ± 0.7	54.6 ± 7.3	57.3 ± 7.3	73.4 ± 4.0	89.0 ± 2.3	86.3 ± 1.9	88.3 ± 0.8	91.5 ± 1.1
MLP-NC	52.2 ± 1.2	52.6 ± 8.9	47.5 ± 6.2	67.1 ± 8.1	87.3 ± 1.3	81.5 ± 2.3	86.5 ± 1.4	92.0 ± 1.3
G-GCN	51.4 ± 0.9	60.8 ± 5.4	46.2 ± 4.8	59.6 ± 5.8	91.4 ± 0.2	86.2 ± 0.2	86.2 ± 0.2	90.6 ± 0.5
LFD-NC	**66.7** ± 1.4	**83.3** ± 3.5	**84.8** ± 5.4	**87.5** ± 6.4	**96.6** ± 0.7	**94.6** ± 1.2	**92.3** ± 1.2	**92.3** ± 1.0

**Table 4 entropy-23-00771-t004:** An ablation study of LFD-NC (%). LTI indicates label-based topology initialization, EPL indicates edge probability learning, DP indicates distance pruning, and TF indicates topology refinement.

LTI	EPL	DP	TF	AUC on Actor	AP on Actor	AUC on Texas	AP on Texas
✓	✓	✓	✓	72.1 ± 1.0	66.7 ± 1.4	88.4 ± 2.8	84.8 ± 5.4
	✓	✓	✓	66.4 ± 1.7	60.1 ± 1.6	86.7 ± 6.7	81.3 ± 8.9
✓	✓	✓		70.7 ± 1.4	65.4 ± 2.1	79.1 ± 8.0	74.1 ± 10.8
✓	✓		✓	60.4 ± 0.8	57.3 ± 0.9	50.9 ± 3.3	58.3 ± 6.5

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
