# Peer review of "Unifying Node Labels, Features, and Distances for Deep Network Completion"

_entropy, 2021, doi:10.3390/e23060771_

Round 1
Reviewer 1 Report
The authors proposed a unified deep graph convolutional network that infers missing edges by leveraging node labels, features, and distances. Although this paper seems readable, this reviewer raises several major comments below.
- The first major concern is about the practicability of the proposed approach. Although the side information of nodes, such as labels, features, and distances, is often available, it is difficult to see a practical application where all of these information are available but there are difficulties in collecting the edges (connectivity information). The authors should make this point clearly by giving practical scenarios. Furthermore, it seems nonsensical to take into account such side information of nodes in performing network completion since we may have no idea about node identification before the network completion task.
1. The second concern is related to the set of shortest distances. Rigorously, if the distance is equal to 1, then it means we know the edges connecting observable nodes and unobservable ones and thus the problem should be re-defined. On the other hand, the set of shortest distances may make the problem trivial as the combination of shortest distances may allow us to infer all the missing edges exactly. The authors should make additional discussions on this issue.
2. In the proposed approach, each type of side information is used in a sequential manner despite a common sense that there are correlations between them. For example, the node labels are highly related with node attributes. In this sense, using node labels only for the initialization seems to be a waste of information (e.g., why don’t we utilize node labels in the edge probability learning step?). Additionally, the authors should also conduct an ablation study to see the effect of each module separately. For example, we can evaluate the performance of the initialization graph from SBM.
3. Equation (11) seems wrong. First, we can only realize a likelihood based on learned parameters. Second, the likelihood cannot be computed from only pairs in $E_Z$ without pairs not in $E_Z$. The authors may refer to page 4 of the paper “The Network Completion Problem: Inferring Missing Nodes and Edges in Networks, SDM 2011” for more details.
Reviewer 2 Report
The paper deals with graph completion, which is a well-known problem in modern computational graph theory and has numerous applications in big (e.g. social) networks analysis.
The main idea of this paper is that incorporating all possible prior information about the considered graph helps to increase the accuracy of the algorithm.
In this paper, the authors represent such data in terms of lengths of the shortest paths and community membership for the nodes of the graph part to be restored.
To solve the problem, the authors propose a neural network approach unifying a number of known algorithms. Although each component of the proposed scheme appears to be known, their combination seems to be novel. Numerical evaluations on several small networks show the supremacy of the proposed algorithm.
Comments
(i) In the intro part of the paper, the authors formulate two graph completion problems. In the former problem, the nodeset of the considered graph is supposed to be known and the goal is to forecast some unknown edges. The latter problem is more difficult since it allows some nodes to be missed in the instance, as well.
Since the proposed algorithm and numerical experiments deal only with the latter problem, mentioning the former seems to be unnecessary.
(ii) The authors claim that their approach is restricted only to undirected unweighted graphs, while it seems to be easily extended to a more wide family of networks, e.g. multigraphs or directed graphs
(iii) The assertion called by the authors Theorem 1 is quite simple (if not evident) and can be presented without any additional proof
(iv) Please, proofread the text, maybe with a help of an English-speaking colleague.
Overall assessment
Generally speaking, I like this paper. In my opinion, its topic is relevant and fits well to the Entropy scope. The results are moderately novel and can be interesting to the specialists. The paper is well structured and easy to follow
I recommend accepting the paper provided the authors agree to make a revision with respect to the aforementioned comments
Round 2
Reviewer 1 Report
Overall, the authors have significantly revised the manuscript. There are some minor comments as follows:
1. Regarding the practicability of the proposed approach, two scenarios described in the revised manuscript are reasonable. However, it would be desirable to describe these two examples more clearly in the revised manuscript. Additionally, the authors should provide a discussion on the difference between the traditional network completion problem defined in “The network completion problem: inferring missing nodes and edges in networks, SDM 2011”, which does not take into account the node identification, and the problem where only edges are missing.
2. The problem under consideration is rather similar to "edgeless" settings below. This reviewer would like to suggest to also cite the following references that attempt to show how to perform link prediction for cold-start nodes having no edges:
[Ref. 1] Y. Hao, X. Cao, Y. Fang, X. Xie, and S. Wang, “Inductive link prediction for nodes having only attribute information,” in Proc. 29th Int. Joint Conf. on Artif. Intell., (IJCAI’20), Virtual Event, Jan. 2020, pp. 1209–1215.
[Ref. 2] Y.-M. Shin, C. Tran, W.-Y. Shin*, and X. Cao, "Edgeless-GNN: Unsupervised inductive edgeless network embedding," arXiv preprint arXiv:2104.05225, 2021.
3. There are typos in lines 230, 287, and 326. Additionally, the authors should carefully check through the manuscript to correct some grammatical errors.
Reviewer 2 Report
As I've mentioned in my previous report, the paper fits well to the journal scope and presents novel results, which can be interesting to the community.
In the revised verision, the authors addressed all my comments carefully. Therefore, I recommend to accept this paper
